# A Complete Review of Automatic Detection, Segmentation, and Quantification of Neovascularization in Optical Coherence Tomography Angiography Images

**DOI:** 10.3390/diagnostics13223407

**Published:** 2023-11-09

**Authors:** Yar Zar Tun, Pakinee Aimmanee

**Affiliations:** School of Information, Computer and Communication Technology (ICT), Sirindhorn International Institute of Technology (SIIT), Thammasat University, Muang, Pathum Thani 12000, Thailand; m6522040051@g.siit.tu.ac.th

**Keywords:** optical coherence tomography angiography (OTCA), neovascularization, diabetic retinopathy (DR), age-related macular degeneration (AMD)

## Abstract

Optical coherence tomography (OCT) is revolutionizing the way we assess eye complications such as diabetic retinopathy (DR) and age-related macular degeneration (AMD). With its ability to provide layer-by-layer information on the retina, OCT enables the early detection of abnormalities emerging underneath the retinal surface. The latest advancement in this field, OCT angiography (OCTA), takes this to the next level by providing detailed vascular information without requiring dye injections. One of the most significant indicators of DR and AMD is neovascularization, the abnormal growth of unhealthy vessels. In this work, the techniques and algorithms used for the automatic detection, classification, and segmentation of neovascularization in OCTA images are explored. From image processing to machine learning and deep learning, works related to automated image analysis of neovascularization are summarized from different points of view. The problems and future work of each method are also discussed.

## 1. Introduction

Optical coherence tomography (OCT) is a non-invasive imaging technique used in ophthalmology and other medical fields to produce high-resolution cross-sectional images of biological tissues. It works by measuring the reflection of light back to the instrument after shining it into the tissue, allowing for in-depth imaging of structures within the tissue. OCT is commonly used for diagnosing and monitoring eye conditions, including age-related macular degeneration and glaucoma [1,2]. Optical coherence tomography angiography (OCTA) is a non-invasive imaging method that provides detailed information about blood flow within retinal and choroidal blood vessels [3]. This method works by analyzing the movement of red blood cells and assessing changes in backscattered light. OCTA has become an essential diagnostic tool for various retinal and choroidal disorders, such as age-related macular degeneration and diabetic retinopathy.

OCTA captures the 3D structure of the retina, and en-face OCTA images are extracted from different retinal layers within the volumetric data. The selection of retinal layers for en-face image extraction varies based on the specific purpose, with no standardized inclusion or exclusion criteria.

Figure 1 provides a layer model and examples of OCTA images at different layers. In this illustration, the retina is divided into four main layers: the superficial retinal layer, the deep retinal layer, the choriocapillaris, and the choroidal layer.

The superficial retinal layer encompasses the top three layers, which include arteries and veins responsible for blood circulation in the deeper retinal layer. The deep retinal layer, located below the superficial layers, contains intermediate and deep capillaries. The superficial layer exhibits larger, more prominently visible blood vessels due to its proximity to the imaging source, while the deep layer displays finer vessels.

The choriocapillaris is a densely interconnected network of capillaries, and a healthy choriocapillaris presents a distinct grainy texture in OCTA images. In contrast, the choroidal layer contains larger vessels when compared to the retinal vasculature. However, these larger choroidal vessels are not as well visualized in OCTA images due to factors like imaging depth limitations, signal attenuation, shadowing effects, the high reflectivity of the layers above, and the specific characteristics of blood flow. Note that the size of the OCTA scan and the focus area within the eye can be adjusted to accommodate the preferences of the ophthalmologist [4].

In ocular diseases, abnormal conditions, such as ischemia, hypoxia, inflammation, and genetic predisposition, can cause neovascularization (NV). The damage to the retina or choroid stimulates the growth of new blood vessels in response to decreased oxygen supply to the affected area [5,6]. OCTA efficiently visualizes neovascular (NV) lesions, especially CNV lesions, beyond the capability of other retinal imaging modalities [7]. Several works [8,9,10,11,12] have reported OCTA’s high NV detection rate across layers.

**Figure 1 diagnostics-13-03407-f001:**
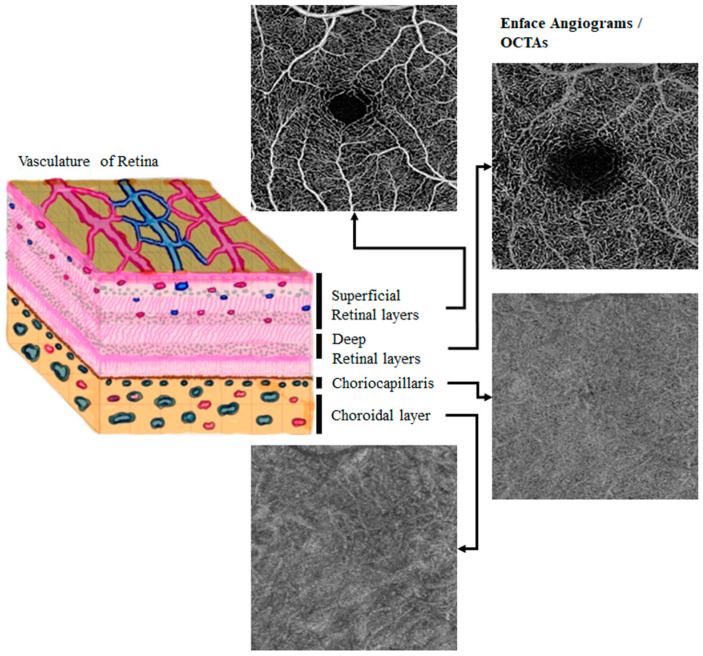
Image showing different layers in OCTA and examples of OCTA images in the superficial retinal and choroidal layers of a normal eye. Shown OCTA images are from the retinal OCTA public datasets [13].

NV can be categorized into retinal neovascularization (RNV) and choroidal neovascularization (CNV) based on the retina’s occurrence layer. RNV arises in the retinal layer extending toward the vitreous cavity, while CNV arises in the choriocapillaris and choroidal layers. RNV is mainly related to diabetic retinopathy (DR), whereas CNV is associated with wet age-related macular degeneration (AMD) [14,15]. Both DR and AMD are untreatable in most patients at severe stages and can cause visual impairment and severe blindness in some cases. NV analysis in the OCT images generally helps prescreen these diseases more effectively than the traditional retinal fundus images because NV can be first detected at layers underneath the retinal surface. The sooner it is discovered, the faster the patient is aware and receives appropriate advice from the doctor to control the condition.

In OCTA images, NV varies in size and pattern. Distinct patterns of CNV previously defined in the literature are the sea fan, medusa or octopus, speckles, and dead tree patterns [14,15,16]. Additionally, a dark halo is usually present around neovascular membranes in CNV images [17]. Figure 2 shows examples of other CNV patterns observed in a public dataset. It’s important to note that the defined patterns rely on each individual’s imagination.

**Figure 2 diagnostics-13-03407-f002:**
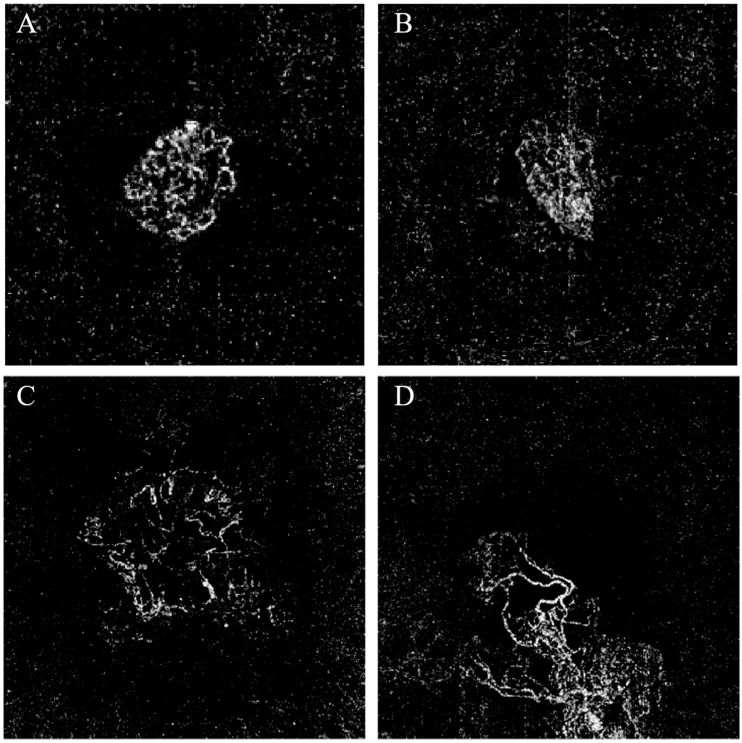
Examples of CNV patterns from a public dataset OCTA-500 [18]: a medusa (**A**), a scatter paint-brush stain (**B**) a moon jelly fish (**C**), and a squid (**D**).

For RNV, primary forms are knots (nodular shape) and small fronds of capillary loops [14,19]. In severe cases, those forms can develop into a more complicated pattern, such as medusa or tangled wools, tight knots or beehives, or a large network of matured thick caliber vessels [20,21,22]. Figure 3 depicts examples of different RNV patterns.

One of the major challenges faced in this research area is the complexity of images. Most techniques struggle with low image quality, as well as noise and background patterns that resemble NV intensity, NV patterns, and scattered artifacts [23,24,25,26,27,28,29]. These issues often lead to over-segmentation and under-segmentation, resulting in low precision and recall, especially in CNV cases [30]. The boundary between NV lesions and other components, such as noise, background, vessels, macular edema, hard exudates, and drusen, is often unclear and can hinder the algorithm from achieving good accuracy.

**Figure 3 diagnostics-13-03407-f003:**
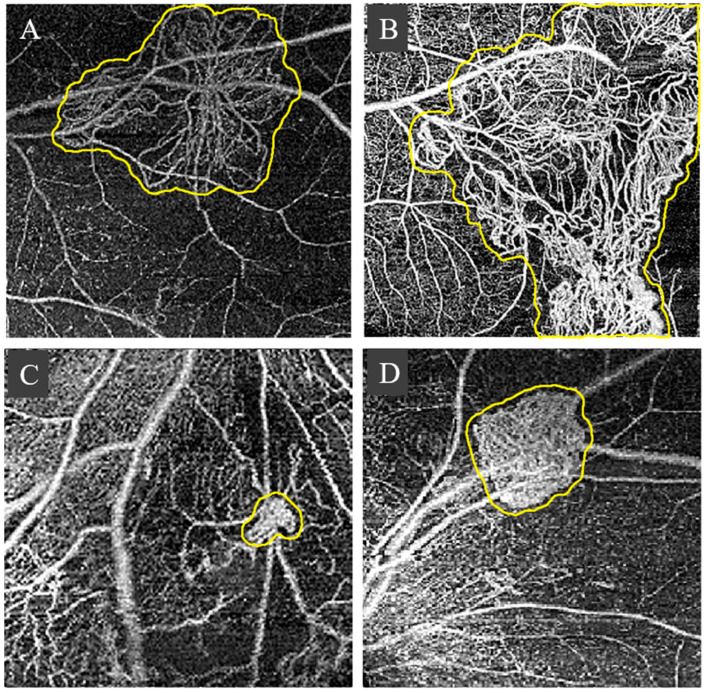
Examples of RNV patterns: medusa (**A**), tangled wools (**B**), a tight knot (**C**), and beehive (**D**). The RNV regions are outlined by yellow lines.

The second challenge in detecting NV lesions using OCTA imaging is the limited availability of such images. This is primarily because OCTA imaging is a relatively new technology. Moreover, OCT images are not extensively researched and have limited datasets and ground truth information available, which further exacerbates the challenge. As a result, most studies reviewed in this regard have limited image samples and small datasets. Due to this, most algorithms fail to learn enough about NV patterns, thus posing a significant challenge in detecting NV lesions using OCTA imaging.

The complexity of NV lesions poses the third obstacle in their diagnosis. While studies have identified common patterns of RNV and CNV, the appearance of NV lesions can vary greatly in size, shape, texture, and location. NV can present itself in different patterns, such as medusa, a sea fan, or a dead tree. The patterns are the results of the vessel crosses. Pattern detection is challenging, as the vessels inside them may not always be separable. Moreover, their sizes are not uniform, as they depend on the severity of a disease.

It is worth noting that early-stage NV lesions especially RNV lesions can appear as similar as intraretinal macular abnormalities (IRMA) because IRMA are precursors to neovascularization [11,31,32,33]. Small knot shapes and small tortuous and loopy NV vessels found in an early NV stage are also mistaken for microaneurysms (MA) [8]. Several clinical research studies have suggested methods for distinguishing CNV and RNV in OCTA images and differentiating them from IRMA, MA, collateral vessels, and telangiectatic vessels [21,22,31,32,33,34,35,36,37,38,39,40,41].

The focus areas of this review are NV analysis from a computer science point of view. The analysis covers automatic detection, segmentation, and quantification. NV types of interest are retinal neovascularization (RNV) and choroidal neovascularization (CNV). RNV is an abnormality sign of DR, while CNV is an abnormality sign of AMD.

A comprehensive search was conducted using Google Scholar, journal publisher websites, and online databases to find published material, including conference proceedings, journal papers, book chapters, and books. We then assessed the relevance of the search results to this review. Specifically, we focused on work that addresses the automatic detection, segmentation, and/or quantification of neovascularization within OCTA datasets.

Studying automatic NV analysis is a hot trend in medical image processing due to the domain’s newness in the retina field and effectiveness in early prescreening of DR and AMD. Existing automatic NV work was conducted solely on RNV or CNV, usually with a single analysis objective on OCTA images. However, no work has provided complete literature reviews across all types of NVs with all aims in NV analysis (detection, localization, segmentation, or quantification), especially in the computer science aspect.

In this review paper, we provide the following:A foundation of knowledge about CNV and RNV;Reviewed techniques in a tabular fashion for the reader to easily digest. The works are sorted by published years. It provides achieved accuracy and other evaluations on different NV analyses for research objectives;For each NV type, reviews and analyses of the used techniques are carried out from three main approaches: deep learning, image processing, and a hybrid of image processing and machine learning, in a tree fashion to conveniently visualize which areas are the most and the least popular;An analysis of the frameworks of techniques, which shows similarities and differences quickly. Detailed methods used by each step in the approach are also summarized in a flowchart fashion;Procedural details of each reviewed method;A summary of the problems with each type of NV that are still open for future work.

The review will be helpful for readers who are computer scientists to learn about the research trend in the overall NV analysis. In addition, it can help them decide their research directions or gain some ideas on improving their research work.

## 2. Reviews on Methods, Used Datasets, and Performance

Since the initial clinical practice of OCTA in 2015, a few works related to NV classification, detection, and quantification have been published. Most of the work found was in the clinical aspect. In a clinical study by Wang et al. [23], OCTA was concluded to have high clinical value in detecting CNV. Another group of researchers, Hirano et al. [42], stated that SS-OCTA [43] slab images had comparable performance in the detection of RNV when compared to fluorescein angiography (FA) images. OCTA images are a future standard in clinical use due to their ability to visualize different biomarkers of complications and their quick and non-invasive nature.

Compared to the fundus and OCT image domains, studies on evaluating NV in OCTA images are scarce. Additionally, most studies on NV detection using OCTA images primarily focus on clinical aspects, such as diagnosing NV or grading its severity, where the NV analysis is manually conducted. Studies related to automatic classification, detection, and segmentation of NV in OCTA images are few and far between. The automation of these tasks is challenging because the morphological characteristics of NV vary from type to type. Table 1 presents a summarized review of the developed algorithms.

For the evaluation of NV detection, standard evaluation schemes were accuracy, the area under the curve (AUC), sensitivity, and specificity. For NV segmentation, the Jaccard similarity (or intersection over union (IOU)), precision, recall, F1 score, and dice coefficient were used. To evaluate NV quantification, the Jaccard similarity was mostly used. The formulae and the definitions of these evaluations can be found in the works by Liu et al. [17], Wang et al. [23], Vali et al. [44], Giavarina [45], Schneider et al. [46], and Carrington et al. [47]. Figure 4 depicts a diagram of methods classified by type, tasks, and methods.

**Table 1 diagnostics-13-03407-t001:** A summary of works related to NV detection and segmentation sorted by years in descending order. AUC is the area under the ROC curve, and CNN is a convolutional neural network.

Authors	Tasks and Purposes	Methods	Datasets	Best Performance
Deshpande et al. (2023) [48]	CNV segmentation and quantification	Gaussian kernel, Frangi filter [49], local adaptive thresholding [50], Mexican hat filter [51], skeletonization	26 manually cropped OCTA images with CNV	Results are evaluated qualitatively; no numerical results are provided for NV detection.
Feng et al. (2023) [52]	CNV segmentation	U-Net with ResNeSt blocks [53] and spatial pyramid pooling [54]	116 OCTA images with neovascular AMD	Accuracy of 98.91% with AUC of 94.76%, a specificity of 99.50%, a sensitivity of 72.71%, IOU of 58.67%, dice of 72.99%, and F1 score of 65.05%.
Wang et al. (2023) [24]	CNV detection, segmentation, and quantification	Dense-Net, U-Net [55], Parallelized-Net, and Res-Net [56]	4701 OCTA images with CNV and 5865 images without CNV	AROC of 97.00%, Sensitivity of 95.00% for CNV diagnosis, and IOU of 66.24%, F1 Score of 78% for CNV segmentation.
Vali et al. (2023) [44]	CNV segmentation and morphological classification	U-Net [55], binarization, color conversion, dilation and region growing, and VGG16 [57]	130 OCTA Images with CNV	A dice coefficient of 90.00%. Accuracy of 84%, 85%, 82%, 81%, and 86%, respectively, for classification of branch, shape, anastomosis and loops, peripheral arcade, and dark halo.
Li et al. (2023) [25]	RNV detection and segmentation	ResNet 101 [56] classifier + 2D V-Net [58] segmentation	109 UW-OCTA training images (35 RNV images and 74 images of other lesions) and 95 UW-OCTA test images	A dice coefficient of 55.66% for RNV segmentation.
Taibouni et al. (2021) [59]	CNV detection for AMD severity grading	VGG19 [57]	391 no AMD, 459 non-neovascular AMD, 548 neovascular AMD images	Accuracy of 89.74%, precision of 96.00%, ROC-AUC of 99.00%, and F1 Score of 84.00%.
Thakoor et al. (2021) [26]	CNV detection for AMD severity grading	3D CNN	97 non-AMD, 169 non-neovascular AMD, and 80 neovascular AMD images	Precision of 40.20% and 42.30%, recall of 70.00% and 72.00%, and F1 Score of 51.10% and 53.30%, respectively for the OCTA + OCT model and the OCTA + OCT + B-scan model.
Wu et al. (2021) [60]	RNV segmentation and quantification	Color space conversion, partial line detection to detect vessels, regional connectivity for vessel extraction, Otsu’s binarization [61] for optimization, morphological operations for noise/artifact reduction	14 eyes with PDR	Results are evaluated qualitatively; no numerical results are provided for NV detection.
Wang et al. (2020) [23]	CNV detection and segmentation for AMD grading	Simplified CNN	1676 images with NV AMD, non-NV AMD-DR	Sensitivity of 100.00% and specificity of 95.00% for CNV diagnosis, Jaccard’s similarity of 88.00% for CNV segmentation with precision of 95.00%, recall and F1 score of 93.00%.
Cheng et al. (2019) [27]	CNV detection and segmentation	Color space transformation, Otsu’s method [61], the majority method for blood vessel extraction, eccentricity features for CNV judgment, and morphological thinning [62] for CNV recognition	17 eyes with CNV	Results were evaluated qualitatively, and no numerical result was provided for NV detection. However, the proposed method can detect NV in only some images.
Taibouni et al. (2019) [28]	CNV segmentation and quantification	Median filter for contrast enhancement, Frangi’s filter [49], or Gabor wavelet filtering [63] for vessel enhancement, hysteresis thresholding via Fuzzy C mean classification [64] for NV vessels detection, and morphological operations for mask generation	54 eyes with neovascular AMD (Type 1 and Type 2 CNV)	Jaccard’s similarity of 87.50%.
Coscas et al., (2018) [65]	CNV segmentation and quantification	Median filter, Phansalkar thresholding [66] for binarization, connected component thresholding for noise removal, density map, custom region growing algorithm for CNV shape detection, morphology analysis for blood flow calculation, and box counting and skeletonization for quantification	104 eyes with neovascular AMD (72 eyes under treatment and 32 eyes under remission)	No direct performance evaluation was reported for segmentation and quantification.
Xue et al. (2018) [29]	CNV segmentation	Gaussian filter for contrast enhancement, Gaussian distribution, DBSCAN method [67] with P Systems [68], and morphological operations for CNV membrane mask	22 eyes with wet AMD	Jaccard’s similarity of 87.20%.
Zhang et al. (2017) [69]	CNV segmentation	Contrast adjustment, morphology analysis, and border detection [70] for NV membrane area extraction	27 eyes from 23 patients with CNV	Correlation coefficients: 89.80% using SS-OCTA images, 82.20% on SD-OCTA images.
Gao et al. (2016) [30]	CNV segmentation and quantification	Saliency method [71] for CNV separation, thresholding for quantification, a level set method [72] for NV vessel segmentation within detected NV membrane, skeletonization [73] for vessel length quantification	9 images of 9 neovascular AMD eyes	Jaccard’s similarity of 69.00%.
Liu et al. (2015) [17]	CNV segmentation and quantification	Gaussian filtering for contrast enhancement, saliency method [71], Laplacian and bilateral filter [74], Otsu’s thresholding [61] for rough CNV region extraction, morphological operations for mask generation	7 images of 7 neovascular AMD eyes	Jaccard’s similarity of 83.40%.

**Figure 4 diagnostics-13-03407-f004:**
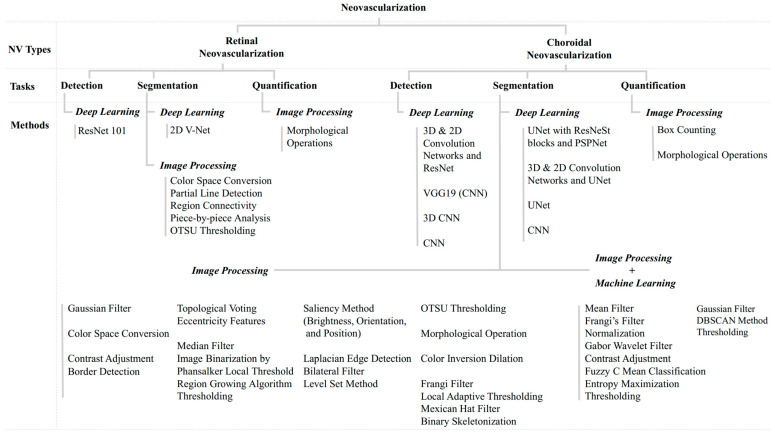
A diagram illustrating a summary of statement-of-the-art methods classified by type, tasks, and methods.

The following can be concluded from Table 1 and Figure 4:The earliest work of automatic NV analysis was CNV segmentation and quantification in 2015.Automatic NV detection was used in several applications, such as DR-stage grading and AMD-type classification.No standard data sets were used in NV detection and segmentation. Most of the data sets used were local images obtained from organizations or the Internet.Most works used NV images from AMD patients. Two groups of researchers used images from DR cases. Algorithm performances depended on data sets and used techniques.The developed algorithms were primarily in CNV detection. Very few works were conducted for RNV.The accuracies of NV detection reported were around 90%. Some work achieved sensitivity as high as 100%. The best Jaccard’s similarity value reported for NV segmentation was 88.00%.For NV detection, only deep learning techniques were explored. These are 2D and 3D convolution networks, ResNet, ResNest 101, VGG19, 2D CNN, and 3D CNN. Modified U-Net with ResNeSt blocks and PSPNet, 2D and 3D convolution networks, U-Net, 2D V-Net, and CNN were employed in the NV segmentation. On the other hand, image-processing methods were largely used for segmentation and quantification.Morphological operations, OTSU binarization, and thresholding were the most popular image-processing methods in segmentation and quantification.

Figure 5 and Figure 6 show frameworks of NV detection, segmentation, and quantification using a combination of image processing and machine learning.

**Figure 5 diagnostics-13-03407-f005:**
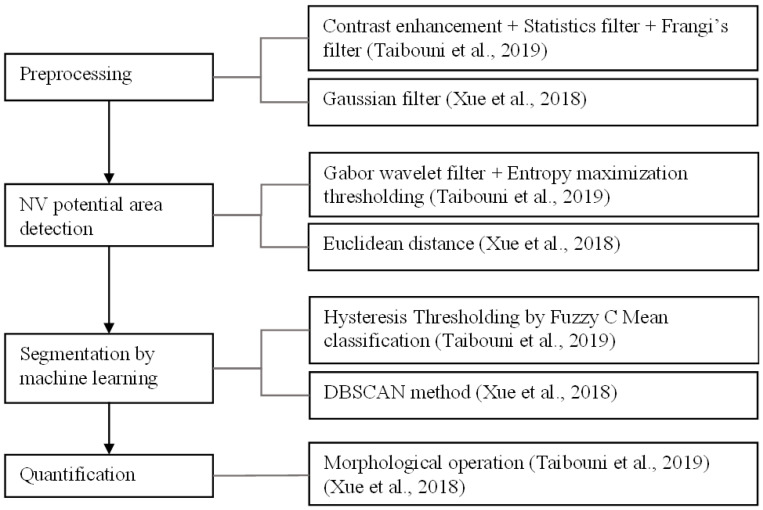
A diagram illustrating the frameworks of state-of-the-art methods used in combination of image processing and machine learning [28,29].

**Figure 6 diagnostics-13-03407-f006:**
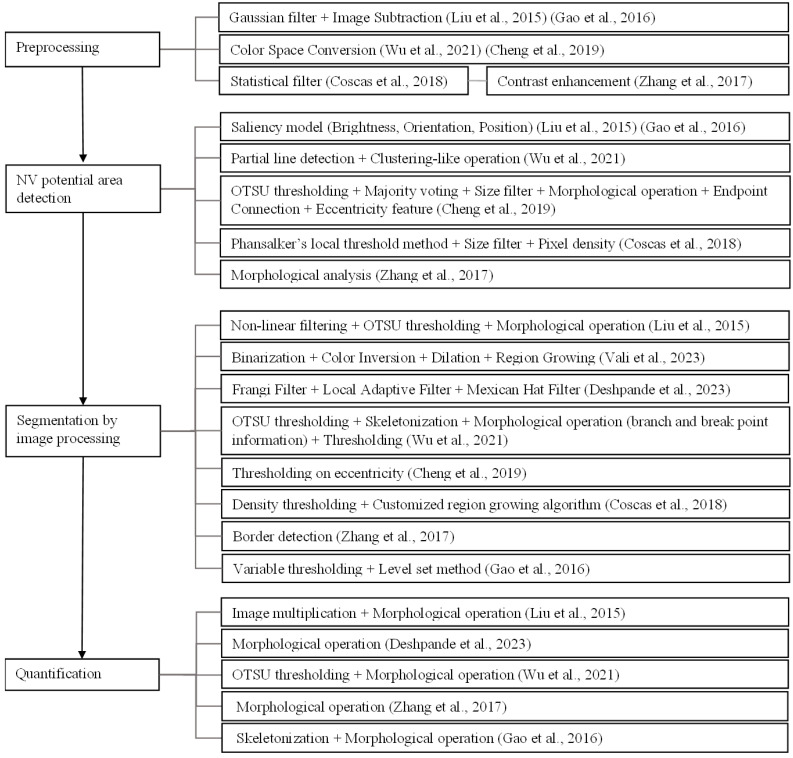
A diagram illustrating the frameworks of state-of-the-art methods using image processing techniques [17,27,30,44,48,60,65,69].

From Figure 5 and Figure 6, the following observations can be made.

In the image-processing approach, the NV detection, segmentation, and quantification frameworks overlap. All tasks usually start by preprocessing and then detecting potential NV areas. Additional segmentation methods in the image-processing approach were applied to obtain the region. For quantification, morphological operations were commonly used to assess the NV regions.Some works used both image processing and machine learning. Image processing techniques were applied to preprocess the image and obtain the features of regions of interest; then, machine learning methods were used on the extracted features to segment the NV boundary.Image subtraction, color space conversion, contrast enhancement, statistical filters, and Gaussian filters were used in the preprocessing.Depending on the target NV shape, texture, or size, different features were extracted using various techniques. The saliency method [71] was used for the CNVs that are salient in the input image in terms of brightness, orientation, and position [17,30] to generate a saliency map. A vascular network extraction method with partial line detection was used for NV of curvilinear structures [60]. Binarization and morphological analysis were applied to the preprocessed image in the works of Cheng et al. [27], Coscas et al. [65], and Zhang et al. [69]. Then, thresholding by the size of connected components was carried out in the works of Cheng et al. [27] and Coscas et al. [65] for the NV that is prominent by size. Eccentricity features were extracted for each candidate region in the work of Cheng et al. [27]. Taibouni et al. [28] used the Frangi filter and Gabor wavelet to classify vessels by thickness to identify the potential NV regions. The distances between non-zero intensity pixels were used as features for a clustering task in the work of Xue et al. [29].The resulting features, feature maps, or candidate regions were further processed to obtain the final ROI. Thresholding followed by morphological operations was the general process for the final candidate extraction. In some cases, the feature map went through some filters before thresholding [17]. In one case, the region growing method was used after obtaining the final ROI to achieve a more accurate segmentation of the CNV region [65]. Hysteresis thresholding via fuzzy C mean classification was employed by Taibouni et al. [28] to obtain the CNV region. DBSCAN clustering [67] with membrane systems [68] was used by Xue et al. [29] in the extraction of NV pixels.Morphological operations were mostly used in the quantification process. Quantifying methods such as box counting and skeletonization were used on the segmented regions to assess the NV region. Different quantifiers such as the vessel’s area, vessel length, vessel width, vessel tortuosity, fractal dimension, bifurcation density, vessel density, lacunary measure, and aspect ratio were reported.For the deep learning approach, different CNN architectures were widely used for segmentation.

Next, we provide detailed settings for deep learning architectures.

### Deep Learning Architectures and Settings

Different settings were applied in full architectures, in a modified architecture, or in a transfer learning approach. VGG19, a type of CNN with 16 convolutional and 3 fully connected trainable layers with 5 max-pooling layers, was customized on some layers by Taibouni et al. [59] in the application of AMD grading. In their work, a rectified linear unit (ReLU) was added as an activation function and a 2 × 2 max pooling layer after convolution layers. Three densely connected layers with 1024, 1024, and 3 nodes were used as the classifier.

Another work by Thakoor et al. (2021) [26] used a CNN architecture on a collection of OCTA images to detect NV for grading AMD severity. The technique used was 3D CNN. The convolution kernels were three-dimensional, and densely connected layers for classification were one-dimensional. OCTA images of five deep retinal layers (the deep retinal, avascular, outer retina choriocapillaris (ORCC), choriocapillaris, and choroid layers) were fed into the 3D CNN. Kernel dimensions of 40 × 40 × 2, 20 × 20 × 2, 10 × 10 × 2, and 5 × 5 × 2 were used in the convolution part. The LeakyReLU activation function and 2 × 2 × 1 max-pooling were used after each convolution layer. A dropout value between 0.05 and 0.5 was used after each max-pooling layer. The He uniform initialization [75] and L1 regularization (Lasso Regression) with a tuning parameter (lambda) to control the bias–variance tradeoff of $10^{-5}$ were used for each convolution block. Two densely connected layers with 64 and 3 nodes were used for the classification. Wang et al. (2020) implemented a CNN architecture to detect the presence of CNV in the given input data. The structural volume of the outer retina and OCTA image sets were fed into the CNN. The max-pooling layers after the convolution layers were replaced with atrous kernels [76,77] to prevent reductions in feature resolution. Atrous kernels with varying dilation rates (1, 2, 4, 8, 16, and 32) were used in the encoder part, and the decoder part was built with a U-Net-like architecture. The CNV membrane probability map was finally produced via SoftMax activation. Thresholding was used on the probability map to detect the presence of the CNV membrane.

Vali et al. [44] developed a U-Net structure to extract CNV masks. The peripheral masks of CNV lesions, such as loops and dark halo, were extracted by image processing. A VGG16 architecture [57] with transfer learning and a small dense layer network was applied to the resulting masks to detect the presence of the morphological features of CNV lesions.

Feng et al. [52] applied a traditional U-Net architecture with the encoders and decoders modified. The encoders and decoders were replaced with ResNeSt blocks [53]. To avoid overlooking the spatial and contextual information of CNV pixels, max-pooling units were replaced with the pyramid scene parsing network (PSPNet) [54]. The sizes of the encoders were 64 × 152 × 152, 256 × 76 × 76, 512 × 38 × 38, 1024 × 20 × 20, and 2048 × 10 × 10. Feature maps in each ResNeSt block were split into two cardinal groups. In each cardinal group, they were divided into two paths, with 1 × 1 and 3 × 3 convolutions each. The resulting features from those paths were fed into a split attention module comprising global pooling, two 1 × 1 convolutions, and a SoftMax activation layer. The resulting features from the modules of the two cardinal groups were merged and fed again into a 1 × 1 convolution block. Inside PSPNet, features were fed into (2 × 2), (3 × 3), (6 × 6), and (9 × 9) pooling kernels, and those were resized and concatenated together with down-sampled features. Then, a 1 × 1 convolution layer followed.

Li et al. [25] were the first group of researchers to apply deep learning techniques to en-face retinal OCTA images to differentiate the RNV lesions from other lesions such as IRMA and non-perfusion areas and to segment RNV lesions. ResNet 101 [56] with default data augmentation, cross-entropy loss, and Adam optimizer is used as the classifier. The RNV segmentation is performed using 2D V-Net [58] with dice loss and Adam optimizer. V-Net [58] is based on U-Net incorporating residual blocks into the network. The training process converged faster by integrating residual linking.

The above works achieved reliable accuracies. However, some false positive and false negative cases were reported in severity grading because of the wide varieties of shapes, sizes, and textures of NV while there were too few data input samples to learn. Furthermore, highly reflective drusens which looked a lot like NVs also caused misdetection. Finally, the CNV vascular network sometimes appeared as speckles like the background. For these reasons, adding more discriminant features and NV images could be helpful for the algorithm to improve its learning ability.

Deep retinal OCTAs, where CNV is commonly found, were subjective to projecting artifacts from superficial retinal layers. Therefore, artifact removal was required in the CNV segmentation and quantification process. The following are techniques used in artifact removal. Coscas et al. [65] removed the artifacts using a program by Zhang et al. [78], which utilized the reverse shadowing effect technique.

In work of Xue et al. [29], artifacts were removed by the reflectance-based projection-resolved OCTA algorithm [79]. Zhang et al. [69] applied their artifact removal algorithm [80]. Gao et al. [30] and Liu et al. [17] subtracted the Gaussian-filtered superficial OCTA from deep OCTA for artifact removal. Cheng et al. [27] managed to detect the CNV in the presence of projection artifacts.

The following are details of works that used deep learning methods for NV segmentation.

Wang et al. [23] built a CNN architecture to segment the detailed CNV vessels in the segmented membrane. The input to and the probability maps from the previous CNN were fed into the model—the same architecture of the prior membrane segmentation CNN was used. But the dilation rates of the atrous kernels [76,77] were reduced to 1, 2, 4, and 8. One zero was inserted between the elements of the 3 × 3 kernel to obtain an atrous kernel of 1. An atrous kernel of 3 was obtained by inserting two zeros between the elements of the 3 × 3 kernel. Other families of deep learning besides CNN, such as Dense-Net, U-Net, Res-Net, and Parallelized-Net, were combined to diagnose and segment CNV lesions. Wang et al. [24] fed outer retinal OCTA and OCT volumes into a 3D subnet. A bridge subnet for 2D feature extraction compressed the resulting features. The extracted features were fed into Res-Net [56] and U-Net [55] for CNV diagnosis and segmentation. The U-Net structure was applied in the work of Vali et al. [44] to segment CNV masks from OCTA images. ReLU activation and 2 × 2 max pooling followed every two 3 × 3 convolutions in the encoding route. In the decoding step, 2 × 2 up-convolution was followed by 3 × 3 convolution with a dropout layer and 3 × 3 convolution with ReLU activation. Finally, 1 × 1 convolution was used to generate a CNV mask.

## 3. Discussions

Many computer science techniques have been applied to automatic NV analysis. Each method has its advantages and disadvantages. The following is a summary of the strategies’ advantages and disadvantages.

Various clustering algorithms were used to segment NV lesions. Xue et al. [29] employed the DBSCAN method [67] with P Systems [68], allowing the segmentation of discontinuous speckle CNV lesions based on distance and connectivity criteria. Gao et al. [30] utilized a level set method with a local intensity clustering criterion to group similar pixels for CNV lesion segmentation, but noise within clusters remained an issue. Taibouni et al. [28] used hysteresis thresholding and the Fuzzy C Means clustering method [64] to establish multiple thresholds, effectively separating NV pixels from noise. The region-growing technique, another widely used clustering method, was employed by Coscas et al. [65] for NV segmentation. This method is most effective when the NV boundary is clearly defined, but less so for regions with medusa or tangled wool patterns, typically characterized by light color or thin vessels.

The algorithms used in image processing are straightforward and easy to comprehend. It usually requires no large OCTA images and still produces satisfactory results. Algorithms that incorporate machine learning and image processing techniques have become more intricate but also more accurate. Despite the complexity and high cost, deep learning algorithms provide the highest accuracies in NV segmentation.

Most developed algorithms that employed deep learning techniques required extensive amounts of data. An average number of 2,037 images were used in each deep learning study. In comparison, only 29 and 38 images were used in the image processing and a hybrid approaches (image processing and machine learning), respectively. In addition, some deep learning methods required special extra inputs to obtain a high accuracy. For instance, in the work of Wang et al. [24], four en-face OCTA and four OCT images of different retinal layers, two retinal thickness maps, OCT volumes, and an OCTA volume were required for each individual entry. Thakoor et al. [26] required a set of six OCTA images, six structural OCT images of different retinal layers, and an OCT B Scan for each single entry. Generally, deep learning algorithms are complex, and expensive in terms of memory and time. They need a large amount of data to generalize the problems.

Feature extraction was popularly used in many studies, especially those that required classification. Most techniques used the intensity feature for CNV detection. Thresholding was commonly used to filter out noise or isolate the area of interest. The saliency method, used by Liu et al. and Gao et al., worked well when CNV was the prominent part of the image. However, it might overlook less obvious CNV regions and focus on noise and artifacts. To address this, OTSU’s binary thresholding was employed to filter out noise vessels.

Variable thresholding and the level set method were more effective than non-linear filtering and OTSU thresholding for segmenting CNV lesions from a noisy background. Zhang et al. [69] enhanced image quality with contrast adjustment and then used intensity thresholding to segment CNV lesions, but this approach worked best for bright, dense CNV lesions on a uniform-intensity background. The method might struggle with speckled CNV lesions or those with unclear borders. The threshold value was typically determined from dataset statistics, which can limit the technique’s repeatability on new images with different CNV characteristics than those in the trained dataset.

Vessel size was crucial for analysis. The Frangi filter [49] was a popular choice for detecting vessels within a specific size range. It was used by Deshpande et al. [48] and Taibouni et al. [28] and was effective when vessels within lesions were distinguishable. Therefore, it was unsuitable for NV lesions with a cotton wool texture or those where vessels were not separable.

Bifurcation points were valuable in certain NV patterns with clearly visible vessels. Wu et al. [60] used a piece-by-piece analysis method to extract parameters from bifurcation points and the length of candidate vascular networks. Thresholding was then applied to these parameters. This method, however, could not handle NV patterns with inseparable vessels because it failed to detect branch points.

The pixel density feature was also considered for inseparable vessels in some NV patterns. Coscas et al. [65] used this feature to generate a pixel density map. Intensity thresholding and region-growing algorithms were then applied to detect the region. This method worked well only if the CNV density was prominent. However, this method did not operate properly when CNV lesions were in speckle form or when high-density artifacts existed. Region growing poses some drawbacks, and one of them is that it often yields over-segmentation or under-segmentation phenomena due to the nonuniform intensity background. Moreover, it is computationally expensive.

In image processing, using eccentricity features can help detect CNVs that were not easily visible based on intensity alone. The eccentricity features distinguished CNV from background noise. Cheng et al. [27] employed this feature successfully. When dealing with artifacts having similar intensity and shape to CNV, along with a highly illuminated, grainy background, the eccentricity-based method outperformed other techniques relying on intensity, density, or saliency. However, it was important to note that CNV could take various forms and shapes, and not all CNV cases exhibited defined eccentricity values.

## 4. Conclusions

Optical coherence tomography (OCT) is a non-invasive imaging method in medical fields like ophthalmology to create detailed cross-sectional images of biological tissues. Optical coherence tomography angiography (OCTA) is a similar non-invasive technique that provides insights into blood flow in retinal and choroidal blood vessels. It is crucial for diagnosing retinal and choroidal disorders like diabetic retinopathy, which can include neovascularization (NV) detectable in OCTA images. However, analyzing NV automatically is challenging due to issues like low image quality, noise, limited OCTA data, and the varied appearances of NV lesions.

NV can be categorized into two types: retinal neovascularization (RNV) and choroidal neovascularization (CNV). RNV patterns are more diverse and challenging to detect and segment than CNV patterns because they resemble the complex background of multi-sized vascular networks. Therefore, RNV research is less extensive than that on CNV.

Research on automatic NV analysis is divided into three main tasks, namely detection, segmentation, and quantification, using methods like image processing, machine learning, and deep learning.

Image processing techniques that have been explored are as follows. For image improvement, the employed techniques were color space conversion, contrast adjustment, and normalization. For noise filtering, the used techniques are Gaussian, bilateral, Mexican hat, and statistics. For component or region detection, techniques were partial line detection, edge/border detection, region connectivity, Gabor wavelet, entropy maximization thresholding, OTSU thresholding, the Frangi filter, local adaptive thresholding, binary skeletonization, morphological operation, box-counting, region connectivity, piece-by-piece analysis, Phansalker thresholding, intensity, and region growing, contrast adjustment, border detection, and topology voting. The machine learning techniques used in conjunction with image processing were entropy maximization, fuzzy C mean classification, and DBSCAN. For deep learning, CNN, UNET, VNet, ResNet101, VGG19, ResNeST, and PSPNet were explored.

The following is a summary of the performance results reported in RNV research. In the segmentation, the highest recorded performance stands at a dice coefficient of 55.56%, achieved through a deep learning approach. Interestingly, there were no reported scores for quantification when using the image processing approach, despite the available literature on the topic.

The reported achievements in CNV research are as follows. For detection, the most outstanding performance is characterized by a sensitivity of 100.00% and a specificity of 95.00%, achieved through deep learning. In segmentation, the leading performance achieved through a deep learning approach includes an 88.00% Jaccard’s similarity, 95.00% precision, and 93.00% recall and F1 score.

Finally, in quantification, the best-reported performance achieves a Jaccard’s similarity of 87.50% and utilizes the image processing approach. The reported problems of automatic NV analysis are misclassification, over and under-segmentation, and inaccurate quantification. The causes of these problems are due to the wide variety of appearance of NV lesions, non-NV artifacts that are too much alike, non-uniform backgrounds, and an inability to detect internal vessels in the region to determine NV using vessel features.

Techniques for improving the visibility of RNV, such as pattern recognition, corner detection, tortuosity detection, vessel clustering, regional contrast adjustment, and vessel convergence, can be explored in future work.

## Data Availability

Not applicable.

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
