# Peer review of "A Complete Review of Automatic Detection, Segmentation, and Quantification of Neovascularization in Optical Coherence Tomography Angiography Images"

_diagnostics, 2023, doi:10.3390/diagnostics13223407_

Round 1

Reviewer 1 Report

Comments and Suggestions for Authors

Reviewer’s Report on the manuscript entitled:

A Complete Review of Automatic Detection, Segmentation, and Quantification of Neovascularization in Optical Coherence Tomography Angiography Images

The authors reviewed methods including image processing and machine learnings utilized for the automatic detection, classification, and segmentation of neovascularization in optical coherence tomography angiography images. I think this is an informative review and can be accepted after some revisions. Below please see my comments.

The literature review needs improvement.

Please search Scopus and Web of Science and search for the most relevant keywords, such as OCTA. For example, I found the following study on OCTA is missing:

https://doi.org/10.1007/s00417-020-04821-6

Figures 2 and 3. It would be nicer to use (a), (b), (c), (d) for the figure panels when referring to them.

Line 281. This section is nice but can be further expanded to highlight the advantages and limitations of the current methods.

Line 305. What about specificity? From a given threshold a model can reach %100 true positive rate but at the same time it can have high false positive rate as well which does not guarantee the model is good. I suggest reporting on AUC here as well.

Line 145. Accuracy, AUC, precision, recall, etc. with application in brain signal processing are also defined and utilized in the following study that may be included here:

https://doi.org/10.1109/JSEN.2023.3237383

Line 167. It would be nice to list all the machine learning and deep learning methods for NV detection. I see in Figure 4 you listed some but it would be nice to list them in a couple lines.  Is there any study utilizing Long Short-Term Memory, ANN, YOLO, ResNet, etc.?

The references format and style need to be according to the journal.

Thank you!

Regards,

Comments on the Quality of English Language

There are style/punctuation/grammar issues that need to be checked and corrected.

Author Response

Reviewer 1

 Comments and Suggestions for Authors

The authors reviewed methods including image processing and machine learnings utilized for the automatic detection, classification, and segmentation of neovascularization in optical coherence tomography angiography images. I think this is an informative review and can be accepted after some revisions. Below please see my comments.

The literature review needs improvement.

  • Please search Scopus and Web of Science and search for the most relevant keywords, such as OCTA. For example, I found the following study on OCTA is missing:

https://doi.org/10.1007/s00417-020-04821-6

Response: We have extensively tried to find more relevant scientific papers, but unfortunately, we only found one more. This new study was added to Table 1.

      Most of the studies, including the one you picked as an example, were focused on the clinical aspects of neovascularization. In those studies, neovascularization was manually detected or segmented, primarily used for diagnosing diseases. However, our work focuses exclusively on algorithms, procedures, and techniques from a computer science aspect for the automatic analysis of neovascularization. This analysis involves the detection, segmentation, and quantification of neovascularization. To make our focus on the computer science aspect more explicit, we have revised and clarified the motivation and focus of our work in the second paragraph of Section 2.

We added how we conducted the search in the first paragraph of Page 5 (Lines 123-127). The added information is as follows.

We conducted a comprehensive search using Google Scholar, journal publisher websites, and online databases to find published material, including conference proceedings, journal papers, book chapters, and books. We then assessed the relevance of the search results to this review. Specifically, we focused on work that addresses the automatic detection, segmentation, and quantification of neovascularization within OCTA datasets.

The following explanation of why the reviews are not many is provided on lines 159-166.

Compared to the fundus and OCT image domains, studies on evaluating NV in OCTA images are scarce. Additionally, most studies on NV detection using OCTA images primarily focus on clinical aspects, such as diagnosing NV or grading its severity, where the NV analysis is manually conducted. Studies related to automatic classification, detection, and segmentation of NV in OCTA images are few and far between. The automation of these tasks is challenging because the morphological characteristics of NV vary from type to type. Table 1 presents a summarized review of the developed algorithms.

  • Figures 2 and 3. It would be nicer to use (a), (b), (c), (d) for the figure panels when referring to them.

Response: Figures 2 and 3 are changed accordingly.

  • Line 281. This section is nice but can be further expanded to highlight the advantages and limitations of the current methods.

Response: We have added advantages and disadvantages of the current methods in Section 3. Discussions.

  • Line 305. What about specificity? From a given threshold a model can reach %100 true positive rate but at the same time it can have high false positive rate as well which does not guarantee the model is good. I suggest reporting on AUC here as well.

Response: Thank you for your kind suggestion. The specificity score has been presented along with the sensitivity score in the conclusion session.

  • Line 145. Accuracy, AUC, precision, recall, etc. with application in brain signal processing are also defined and utilized in the following study that may be included here:

https://doi.org/10.1109/JSEN.2023.3237383

Response: For the evaluation’s definitions, we cited a comprehensive set of relevant sources for the evaluation's definition in our automatic NV analysis application. Adding more definitions from another field can confuse the reader, so we apologize for not including the recommended work in our paper.

  • Line 167. It would be nice to list all the machine learning and deep learning methods for NV detection. I see in Figure 4 you listed some but it would be nice to list them in a couple lines.  Is there any study utilizing Long Short-Term Memory, ANN, YOLO, ResNet, etc.?

Response: We have listed all the machine learning methods used in NV detection and segmentation on Line 245-248 in section 2, and deep learning methods on Lines 195-199 in section 2.

  • The reference format and style need to be according to the journal.

Response: All the references and citations have been fixed according to the Journal’s format.

Reviewer 2 Report

Comments and Suggestions for Authors

The authors present the revision regarding automatic detection, segmentation, and quantification of neovascularization in angiography images. A suitable presentation is observed; however, it is necessary to review some aspects to improve the paper.

1. It is recommended to review the format used to cite bibliographic references according to MDPI guidelines.

2. Please, clarify the aspects to be evidenced by this review (the review’s contribution). How does the paper differ from similar reviews?

3. It is recommended to expand the explanation in Table 1.

4. Consider the expansion of explanation of Figure 1.

5. Please clarify the methodology used to perform the review. The most widely used is PRISMA (Preferred Reporting Items for Systematic Reviews and Meta-analysis).

6. The most appropriate title for section 3 could be: ''Discussion''.

7. The description of future work should be in the conclusions.

8. The conclusions must be expanded according to the contributions that the authors want to achieve.

9. A general revision of the article is suggested to improve the writing.

10. Turnitin reports a 40% of overall similarity; therefore, please review and correct this aspect.

Comments on the Quality of English Language

Minor editing of English language required.

Reviewer 3 Report

Comments and Suggestions for Authors

1)      Line 485, correct &

2)      Line 12 “allows us”, line 15 “we explore” line 17 “we summarize”, (etc): when writing a scientific article, it must be written in the third person, or use passive voice.

3)      The authors present several figures already published in the literature, which is not recommended in a scientific paper, as the images must all be original to not infringe copyright.

4)      The style for presenting reviews “in a tabular fashion” is not common and most researchers expect to find a review in the traditional format.

5)      Page 5 has only 3 lines; authors should be concerned with the presentation of the document, avoiding this situation.

6)      Table 1 is more readable if it has vertical and horizontal lines, since the column spacing is very small and the reader may be unsure whether the line continues straight ahead or changes lines.

7)      Figures 5 and 6 occupy 1 page and another half page and the discussion presented corresponds to just a few lines.

8)      The authors state in the title “Complete Review” but there are no complete reviews and in the case of this work, in table 1, it presents only 3 works from 2023. A search on the MDPI website in the journal Diagnostics, only in the year 2023, presents 77 articles . Why did the authors only include 3 articles from the year 2023?

9)      Chapter 3 occupies approximately 20 lines and chapter 4 Conclusions occupies only 6 lines, unacceptable in a scientific paper intended to be published in an international journal.

10)   A review article begins by summarizing the methods developed by other authors and classifying these approaches into method families. This approach does not occur in the present paper, the authors limiting themselves to summarizing the work and presenting them in the form of tables and figures, not seeking to associate the methods by similarity, nor mentioning the advantages and disadvantages of each group of methods that are similar.

Author Response

Reviewer 3

Comments and Suggestions for Authors

  • Line 485, correct &

Response: All the references and citations have been fixed according to the Journal’s format.

2)      Line 12 “allows us”, line 15 “we explore” line 17 “we summarize”, (etc): when writing a scientific article, it must be written in the third person, or use passive voice.

Response: Thank you for your comment, and we appreciate your feedback. We understand that our writing style, particularly the use of active voice, may not align with your preference. However, we hold a different perspective on this matter. In scientific articles, the choice between active and passive voice, as well as the use of the first person, can vary depending on the writing style and the specific focus of a sentence. We believe that active voice can contribute to brevity, naturalness, and clarity in certain instances. Additionally, it is not uncommon in journal papers for authors to use "We" to represent themselves, which is an accepted practice. In our review paper, we employed active voice in select sentences when it was better suited for describing our actions.

3)      The authors present several figures already published in the literature, which is not recommended in a scientific paper, as the images must all be original to not infringe copyright.

Response:   In our original manuscript, we obtained permission to use certain images from Dr. Karacorlu, who was one of the authors of the paper titled "Membrane patterns in eyes with choroidal neovascularization on optical coherence tomography angiography." We duly acknowledged this approval and the image source. However, in an abundance of caution to ensure compliance with the journal's policies, we chose to source our own images from publicly available datasets for the revised manuscript.

       In the revised manuscript, the OCTA examples presented in Figure 1 have been sourced from "Retinal OCT and OCTA data (preprocessed)" [https://www.kaggle.com/datasets/cnzakimuena/retinal-oct-and-octa-data-4], while those in Figure 2 are sourced from "OCTA-500" [https://ieee-dataport.org/open-access/octa-500]. The remaining OCTA images used in other figures are from Thammasat Chalermprakiat Hospital.

4)      The style for presenting reviews “in a tabular fashion” is not common and most researchers expect to find a review in the traditional format.

Response:   We are sorry to hear that you disliked our approach. However, we respectfully disagree with your opinion. Presenting reviews in a tabular format enables readers to visualize and compare studies more effectively, ultimately leading to better engagement. This format is widely used in review papers, and it's not necessary to stick to a traditional style. The most important thing is to ensure that readers can easily understand and apply the paper's contributions.

5)      Page 5 has only 3 lines; authors should be concerned with the presentation of the document, avoiding this situation.

Response: This situation has been fixed after making some adjustments.

6)      Table 1 is more readable if it has vertical and horizontal lines, since the column spacing is very small and the reader may be unsure whether the line continues straight ahead or changes lines.

Response:  Adding vertical lines to enhance readability made it harder to comprehend. Only horizontal lines were added based on the suggestion.

7)      Figures 5 and 6 occupy 1 page and another half page and the discussion presented corresponds to just a few lines.

     Response: Thank you for you feedback. The discussion section has been extended by over one page to cover pros, cons, and limitations, and interesting observations (Lines 212-256) of Figures 5 and 6 have been added after these figures.

8)      The authors state in the title “Complete Review” but there are no complete reviews and in the case of this work, in table 1, it presents only 3 works from 2023. A search on the MDPI website in the journal Diagnostics, only in the year 2023, presents 77 articles. Why did the authors only include 3 articles from the year 2023?

      Response:  We have extensively tried to find more relevant scientific papers, but unfortunately, we only found one more recent study. This new study was added to Table 1.

      Most of the found studies, including all 77 articles that you mentioned, were focused on the clinical aspects of neovascularization. In those studies, neovascularization was manually detected or segmented, primarily used for diagnosing diseases. However, our work focuses exclusively on algorithms, procedures, and techniques from a computer science aspect for automatically analyzing neovascularization. This analysis involves the detection, segmentation, and quantification of neovascularization. To make our focus on the computer science aspect more explicit, we have revised and clarified the motivation and focus of our work in the second paragraph of Section 2.

       We added how we conducted the search in the first paragraph of Page 5 (Lines 123-127). The added information is as follows.

       We conducted a comprehensive search using Google Scholar, journal publisher websites, and online databases to find published material, including conference proceedings, journal papers, book chapters, and books. We then assessed the relevance of the search results to this review. Specifically, we focused on work that addresses the automatic detection, segmentation, and quantification of neovascularization within OCTA datasets.

      The following explanation of why the reviews are not many is provided on lines 159-166.

       Compared to the fundus and OCT image domains, studies on evaluating NV in OCTA images are scarce. Additionally, most studies on NV detection using OCTA images primarily focus on clinical aspects, such as diagnosing NV or grading its severity, where the NV analysis is manually conducted. Studies related to automatic classification, detection, and segmentation of NV in OCTA images are few and far between. The automation of these tasks is challenging because the morphological characteristics of NV vary from type to type. Table 1 presents a summarized review of the developed algorithms.

9)      Chapter 3 occupies approximately 20 lines and Chapter 4 Conclusions occupies only 6 lines, unacceptable in a scientific paper intended to be published in an international journal.

Response: We have included additional information regarding the pros and cons, as well as limitations, in the discussions presented in Section 3. Furthermore, we have enhanced the content of the conclusion in Section 4.

10)   A review article begins by summarizing the methods developed by other authors and classifying these approaches into method families. This approach does not occur in the present paper, the authors limiting themselves to summarizing the work and presenting them in the form of tables and figures, not seeking to associate the methods by similarity, nor mentioning the advantages and disadvantages of each group of methods that are similar.

Response: We have illustrated the similarity between the methods using flow charts which are presented in Figures 5 and 6. The detailed explanations for these figures can be found in lines 212-256. Additionally, we have included the advantages and disadvantages of each technique used in similar methods in Section 3, which is dedicated to discussion.

Round 2

Reviewer 3 Report

Comments and Suggestions for Authors

Analyzing the authors' responses, it can be seen that the authors are very convinced of their ideas, not applying all of this reviewer's recommendations. Although it is not mandatory to follow the reviewers' recommendations, the final quality of the article would be higher if they had followed all the recommendations and not just some recommendations.